# Persistent COVID-19 symptoms in community-living older adults from the Canadian Longitudinal Study on Aging (CLSA)

Lauren E. Griffith[1,2,3], Marla Beauchamp[4,2,3], Jacqueline McMillan [5], Sayem Borhan[1], Urun Erbas Oz[1], Christina Wolfson[6,7,8], Susan Kirkland[9], Nicole E. Basta[6], Mary Thompson[10], Parminder Raina [1,2,3✉] & on behalf of the Canadian Longitudinal Study on Aging (CLSA) Team*

## Abstract

**Background** Symptom persistence in non-hospitalized COVID-19 patients, also known as Long COVID or Post-acute Sequelae of COVID-19, is not well characterized or understood, and few studies have included non-COVID-19 control groups.

**Methods** We used data from a cross-sectional COVID-19 questionnaire (September-December 2020) linked to baseline (2011–2015) and follow-up (2015–2018) data from a population-based cohort including 23,757 adults 50+ years to examine how age, sex, and pre-pandemic physical, psychological, social, and functional health were related to the severity and persistence of 23 COVID-19-related symptoms experienced between March 2020 and questionnaire completion.

**Results** The most common symptoms are fatigue, dry cough, muscle/joint pain, sore throat, headache, and runny nose; reported by over 25% of participant who had ($n = 121$) or did not have ($n = 23,636$) COVID-19 during the study period. The cumulative incidence of moderate/severe symptoms in people with COVID-19 is more than double that reported by people without COVID-19, with the absolute difference ranging from 16.8% (runny nose) to 37.8% (fatigue). Approximately 60% of male and 73% of female participants with COVID-19 report at least one symptom persisting >1 month. Persistence >1 month is higher in females (aIRR = 1.68; 95% CI: 1.03, 2.73) and those with multimorbidity (aIRR = 1.90; 95% CI: 1.02, 3.49); persistence >3 months decreases by 15% with each unit increase in subjective social status after adjusting for age, sex and multimorbidity.

**Conclusions** Many people living in the community who were not hospitalized for COVID-19 still experience symptoms 1- and 3-months post infection. These data suggest that additional supports, for example access to rehabilitative care, are needed to help some individuals fully recover.

## Plain Language Summary

Some people who develop COVID-19 experience persistence of symptoms. Here, we aimed to understand the factors associated with the severity and persistence of these symptoms in adults 50 years and older living in the community who had COVID-19. Using information provided by 23,757 participants from across Canada we compared the symptoms between those who had COVID-19 and those who did not. The number and severity of symptoms in participants who had COVID-19 was beyond what would be expected due to other causes. Over two-thirds of participants who had COVID-19 reported symptoms persisting for more than one month, and over half of the participants more than three months. Symptom persistence was higher in females, those with multiple chronic conditions, and lower perceived social status. This suggests that a substantial proportion of people who were not hospitalized for COVID-19 may require further healthcare assistance.

[1] Department of Health Research Methods, Evidence, and Impact, McMaster University, Hamilton, ON, Canada. [2] Labarge Centre for Mobility in Aging, McMaster University, Hamilton, ON, Canada. [3] McMaster Institute for Research on Aging, McMaster University, Hamilton, ON, Canada. [4] School of Rehabilitation Science, McMaster University, Hamilton, ON, Canada. [5] Division of Geriatric Medicine, Department of Medicine, Cumming School of Medicine, University of Calgary, Calgary, AB, Canada. [6] Department of Epidemiology, Biostatistics, and Occupational Health, McGill University, Montreal, QC, Canada. [7] Department of Medicine, McGill University, Montreal, QC, Canada. [8] Research Institute of the McGill University Health Centre, Montreal, QC, Canada. [9] Department of Community Health & Epidemiology, Dalhousie University, Halifax, NS, Canada. [10] Department of Statistics and Actuarial Science, University of Waterloo, Waterloo, ON, Canada. *A list of authors and their affiliations appears at the end of the paper. ✉email: praina@mcmaster.ca

In addition to the immediate health consequences of coronavirus disease 2019 (COVID-19), it is becoming increasingly evident that many people experience persistent symptoms for more than 4 weeks, often termed "long COVID"[1]. On February 25, 2021, the World Health Organization urged national authorities to prioritize the understanding the long-term consequences of COVID-19 infections[2]. Research on how best to identify those at greatest risk of long COVID is urgently required[3], especially in older adults[4].

To date, most studies of symptom persistence have been conducted amongst individuals requiring hospitalization for COVID-19[5–8]. The largest study conducted by Huang et al[9]. found that 76% of patients reported at least one symptom 6 months after symptom onset, the most common being fatigue, weakness, sleep difficulties, and anxiety/depression. While information on survivors of serious COVID-19 illness is critically important, millions have managed COVID-19 in the community without being hospitalized[10,11]. Of the few studies examining symptom persistence in both hospitalized and non-hospitalized adults, most use convenience sampling and/or have low response rates limiting the generalizability of results[1,12–15], or focus on special subgroups (e.g., COVID cases with severe symptoms)[16–18] Those using administrative data[19,20] include well defined populations, but include limited information on socio-demographic and contextual variables that can help better understand symptom persistence. Importantly, only one study includes a control group[20] making it difficult to determine to what extent the long-term symptoms reported by people with COVID-19 can be directly attributed to SARS CoV-2 infection. Furthermore, only one other long-COVID study is embedded in an existing population-based cohort. Matta et al. examine symptoms persisting ≥8 weeks among French adults[21] using both information on the presence of antibodies to SARS-CoV-2 virus and self-report COVID-19 status, but does not examine other relevant definitions of symptom persistence, such as post-acute (>1 month) and chronic (>3 months) as suggested by Greenhalgh et al.[22].

To help address these research gaps and better understand community-based symptom persistence, we analyze data from a COVID-19 questionnaire launched in an existing population-based cohort, the Canadian Longitudinal Study on Aging (CLSA). In this study, we estimate the cumulative incidence of symptoms reported during the pandemic in participants with and without a COVID-19 diagnosis by age and sex. In those with COVID-19, we further examine symptom persistence and the factors associated with the number of persistent symptoms.

## Methods
**Study design/setting**. Globally, the CLSA is one of the largest and most comprehensive research platforms examining health and aging[23]. Recruitment and CLSA baseline data were collected on 51,338 participants in 2011–2015, follow-up 1 (FUP1) was completed on 48,893 participants (95% retention) in 2015–2018, and FUP2 was completed in mid-2021. The CLSA COVID-19 study was launched on April 15, 2020 and comprised a COVID-19 baseline questionnaire, weekly/biweekly/monthly symptoms questionnaires, and an Exit questionnaire. The Exit questionnaire (September 29–December 29, 2020) captured information on COVID-19 symptom persistence. This analysis uses data from the CLSA baseline, FUP1, and the COVID-19 questionnaires. The study was approved by the Hamilton Integrated Research Ethics Board and by the research ethics boards of all the participating institutions across Canada (Supplementary Table 1). Written informed consent was obtained all participants. Participants had the study and data collection procedures explained to them and had the opportunity to seek clarification before consenting.

**Participants**. CLSA participants were community-living adults aged 45–85 years residing in the ten provinces of Canada at recruitment (2010–2015). Full-time armed forces members, people living on First Nations reserves, residing in institutions, unable to respond in English or French, or with cognitive impairment were excluded. Of 51,338 CLSA participants 42,700 were invited to take part in the CLSA COVID-19 study. The 8633 excluded comprised individuals who died ($n = 2500$), had withdrawn prior to FUP2 ($n = 3406$), required a proxy ($n = 318$), or were unreachable for administrative reasons ($n = 2414$). Participants with email addresses were invited to participate via a Web questionnaire ($n = 34,498$) and all others completed telephone interviews ($n = 8202$). During the CLSA COVID-19 Study recruitment process, an additional 166 participants were identified as deceased and 23 as needing a proxy to arrive at the overall eligible sample of 42,511, of which 28,559 (67.2%) agreed to participate in the study and 24,114 (56.7%) completed the Exit questionnaire (Supplementary Table 2a). Compared to those not participating ($n = 18,343$), those completing the Exit questionnaire ($n = 24,114$) were more likely to be aged 65-74 (34.7% vs. 33.1%), less likely to be <55 (13.2% vs. 18.20%), and less likely to be a current smoker (5.8% vs. 8.9%). (Supplementary Table 2b). Our analyses include participants who reported a positive test for or a physician diagnosis of COVID-19 (COVID-19 group) or did not (non-COVID-19 group). Participants who reported it was "very likely" they had COVID-19 but did not have a positive test or diagnosis ($n = 357$) were excluded from the primary analyses. A subgroup of these participants were included in sensitivity analyses (see "Statistical analysis").

**Variables**. All CLSA questionnaires are available on the CLSA website (https://www.clsa-elcv.ca/data-collection).

*COVID-19 status/symptoms*. Participants reporting a positive COVID-19 test or were told they had COVID-19 by a healthcare professional were considered "COVID-19 positive." We included those told by a healthcare professional as many people were not tested early in the pandemic[10]. Participants reporting they "very likely" had COVID-19 but did not report a positive test or a physician diagnosis were excluded from analyses ($n = 357$). All Participants were asked if they experienced any of 23 COVID-related symptoms since March 1, 2020 and if so to rate the symptom as mild, moderate, or severe. There was also the option for participants to report "other" symptoms beyond the 23 included in the survey. Participants with COVID-19 were further asked how long the symptom persisted (2 weeks or less, >2 weeks, >1 month, >2 months, >3 months, or ongoing). We considered two persistent symptom definitions based on the Greenhalgh et al[22]. classification of post-acute (>1 month) and chronic (>3 months) COVID-19.

Symptom status was also collected at the baseline, 2 bi-weekly, and 3 monthly questionnaires. For each of the 23 symptoms we summed the number of times the symptom was reported across the six questionnaires (n_sym). Thus, for participants responding to all six questionnaires n_sym was between 0 and 6. To assess the validity of self-reported symptom persistence in the COVID-19 positive participants at the Exit interview, we compared the mean number of symptoms (n_sym) for those indicating the symptom was persistent (>1 month) to those who did not. For each symptom, n_sym was significantly higher in the persistent symptom group; with an average of 2.4 more reports of the symptom over the six follow-up questionnaires (Supplementary Table 3). When persistence status was missing ($n = 52$ (2.1%)), symptoms reported on the previous six questionnaires were used to impute the status when possible. For example, if a participant

**Table 1 Demographic and pre-pandemic physical, psychological and social health and function data for 23,757 participants of the CLSA COVID-19 Exit interview, overall and by COVID-19 status.**

| Characteristic[a,b] | Overall[c] (n = 23,757) | | No COVID-19 (n = 23,636) | | COVID-19[d] (n = 121) | | Standardized difference | p value |
|---|---|---|---|---|---|---|---|---|
| Female (n, %) | 12,629 | 53.16 | 12,555 | 53.12 | 74 | 61.16 | 0.16 | 0.094 |
| Age group (n, %) | | | | | | | | |
| <65 | 7892 | 33.22 | 7850 | 33.21 | 42 | 34.71 | 0.03 | 0.691 |
| 65–74 | 8735 | 36.77 | 8688 | 36.76 | 47 | 38.84 | 0.04 | |
| 75+ | 7130 | 30.01 | 7098 | 30.03 | 32 | 26.45 | −0.08 | |
| Number of chronic conditions (n, %) | | | | | | | | |
| 0 or 1 | 7448 | 32.63 | 7422 | 32.69 | 26 | 21.85 | −0.25 | 0.016 |
| 2+ | 15,375 | 67.37 | 15,282 | 67.31 | 93 | 78.15 | 0.25 | |
| Depression or anxiety (n, %) | | | | | | | | |
| Yes | 4842 | 20.74 | 4803 | 20.68 | 39 | 32.50 | 0.27 | 0.002 |
| Subjective social status | | | | | | | | |
| Mean (SD) | 6.45 | (1.83) | 6.45 | (1.83) | 6.10 | (1.98) | 0.18 | 0.042 |
| Mobility ADL limitation (n, %) | | | | | | | | |
| Yes | 4297 | 18.22 | 4266 | 18.18 | 31 | 26.27 | 0.20 | 0.031 |

[a]Pre-pandemic health and function.
[b]Participants had missing data for number of chronic conditions 934 (3.9%), depression or anxiety 413 (1.7%), social subjective status 1,015 (4.3%), and mobility ADL limitations 171 (0.7%).
[c]Participants reporting they "very likely" had COVID-19 but did not report a positive test or a physician diagnosis were excluded from analyses (n = 357).
[d]COVID-19 status based on self-reported positive COVID-19 test or physician diagnosis.

with missing persistence data reported a symptom at the Exit but did report the symptom on any of the six preceding questionnaires, their symptoms status was imputed "not persistent" and if a participant reported the symptom at all three monthly questionnaires preceding the Exit interview their symptom status was imputed "persistent" for both definitions (>1 month and >3 months). As a sensitivity analysis we also used multiple imputations for missing persistence information.

*Covariates.* All analyses included age (50–65, 65–74, and ≥75 years) and sex. Regression analyses also included indicators of physical, psychological, and social health and function taken from the participants' FUP1 data (pre-pandemic). Multimorbidity was based on the number of chronic conditions from 10 disease categories (musculoskeletal, respiratory, cardiovascular, endocrine-metabolic, neurological, gastrointestinal, genitourinary, ophthalmologic, renal, and cancer) and categorized into "0 or 1" vs. 2+. Self-reported depression and/or anxiety was considered an indicator of psychological health. Relative social standing was operationalized as a continuous variable using MacArthur social ladder scale[24]. Participants were asked to place themselves on a ladder where the steps represented where people stand in their communities; the top step (10) represented the highest standing and bottom step (1) represented the lowest standing[25]. Participants needing help with any of the following mobility-related activities of daily living: dressing, walking, getting to bed, bathing and toileting, shopping, or housework they were considered to have a functional limitation[26].

**Statistics and reproducibility**. Descriptive statistics were calculated as mean (standard deviation (SD)) for continuous variables and n (%) for categorical variables; standardized differences and two-sided p-values were reported comparing those with and without COVID-19. We graphically displayed the proportion of participants with and without COVID-19 experiencing mild and moderate/severe symptoms for each of the 23 symptoms ordered by overall symptom cumulative incidence in the COVID-19 group. Because multimorbidity is strongly associated with symptoms and was more common in the COVID-19 group, we conducted a sensitivity analysis in which we used direct standardization in the COVID-19 group to reflect the age, sex, and

multimorbidity distribution of the non-COVID-19 group. We further stratified these figures by sex and age and qualitatively examined how patterns varied across sub-groups.

We graphically summarized the 10 most prevalent persistent symptoms (>1 month and >3 months) reported as moderate/severe by the COVID-19 group. We then calculated the overall number of persistent symptoms. Univariate and multivariable negative binomial regression was used to identify covariates associated with the risk of persistent symptoms. Negative binomial regression was chosen rather than Poisson regression because the data were over-dispersed. Data were analyzed separately for the two persistence definitions (>1 month and >3 months). Age and sex were forced into each multivariable model and then a stepwise procedure was used to identity additional indicators of physical (multimorbidity), psychological (anxiety/depression), social (subjective social status) health, and physical function (mobility-related activities of daily living limitations) were independently associated with persistence using a $p < 0.1$ as criteria for inclusion. Model fit was assessed using the deviance goodness of fit test. Incidence rate ratios (IRR) and 95% confidence intervals (CIs) were reported. As a sensitivity analysis multiple imputation using Classification and Regression Trees method was conducted to create 5 datasets used to estimate the IRRs for the final models. Finally, for 15 participants indicating they "very likely" had COVID-19 but did not report a positive test or a physician diagnosis (currently excluded from our analysis), we had seroprevalence results indicating that they had COVID-19 antibodies and were not vaccinated (see details in Supplementary Table 7). Therefore, as a final sensitivity analysis, we included these participants in the persistent symptom regression analysis. All analyses were conducted using R version 4.1.0[27] or SAS Enterprise Guide 8.3 (https://www.sas.com/content/dam/SAS/en_us/doc/factsheet/sas-enterprise-guide-101431.pdf).

**Reporting summary**. Further information on research design is available in the Nature Portfolio Reporting Summary linked to this article.

## Results

**Participants**. Table 1 displays the characteristics of 23,757 Exit interview respondents overall and by COVID-19 status. The

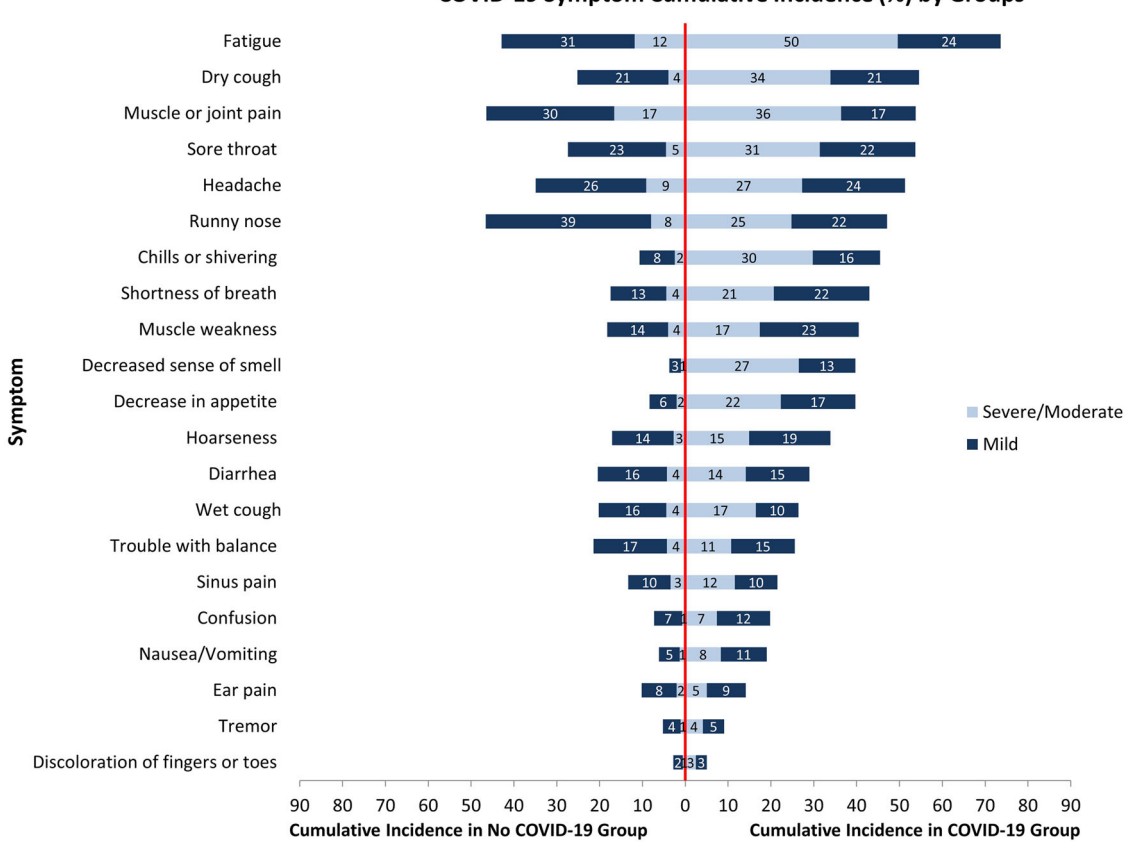

**Fig. 1 Cumulative incidence of mild and moderate/severe symptoms.** Cumulative incidence percent of mild (dark blue) and moderate/severe (light blue) symptoms since March 2020 reported by CLSA COVID-19 study participant who had COVID-19 ($n = 121$) and who did not have COVID-19 ($n = 23,636$). Data to reproduce this figure are included in Supplementary Data 1.

sample comprised 53.16% females and approximately one-third were <65, 65–74, and 75+ years old. Compared to the non-COVID-19 group ($n = 23,636$), those reporting COVID-19 ($n = 121$) had more multimorbidity (78.15% vs. 67.31%), depression/anxiety (32.50% vs. 20.68%), mobility limitations; 26.27% vs. 18.18%), and lower subjective social status (6.10 vs. 6.45) but there were no significant differences in sex or age. Of the COVID-19 participants, 14 reported visiting the emergency department and 7 (5.8%) reported a hospital admission, of whom 3 (2.5%) reported an ICU stay.

Figure 1 displays the proportion of participants with and without COVID-19 reporting 23 symptoms since the beginning of the pandemic. Over 25% of both groups reported fatigue, dry cough, muscle or joint pain, sore throat, headache, and runny nose. While the overall cumulative incidence of many symptoms was high in both groups, the proportion reporting moderate/severe symptoms was consistently higher in the COVID-19 group with over 35% reporting moderate/severe fatigue and muscle or joint pain. Symptoms in the COVID-19 group were only slightly attenuated after direct standardization (Supplementary Fig. 1). Other symptoms reported by participants (beyond the 23 listed) are summarized in Supplementary Table 4.

Female participants with COVID-19 tended to report more moderate/severe symptoms than males (Fig. 2a, b). This was most evident for fatigue (52.7% vs. 44.7%), muscle and joint pain (40.5% vs. 30.4%), headache (32.4% vs. 19.2%), and decreased sense of smell (33.78% vs. 14.9%). Generally, participants with COVID-19 who were <65 years old reported a higher cumulative incidence of symptoms than those of age 65–74 and 75+ years; however, a higher proportion of the symptoms reported in the

75+ COVID-19 group were moderate/severe (Supplementary Fig. 2a–c).

For participants with COVID-19, persistence of >1 month and >3 months was reported for both mild and moderate/severe symptoms, but persistence patterns varied by symptom (Fig. 3). Over 15% reported moderate/severe fatigue, dry cough, and decreased sense of smell, and over 10% reported moderate/severe shortness of breath, muscle or joint pain, or runny nose for >1 month; over 10% reported mild fatigue for >3 months. In addition, over half of the respondents reporting moderate/severe shortness or breath, decreased sense of smell, trouble with balance, and wet cough indicated the symptom lasted >1 month and over a third indicated they lasted >3 months.

Approximately 60% of male participants reported at least one symptom persisting >1 month compared to 75% of female participants, with over 10% of females reporting 8 or more persistent symptoms compared to 4% of males (Supplementary Fig. 3a). More than 50% of male and 40% of female participants reported no persistent symptoms >3 months, while over 10% of females compared to 2% of males reported 5 or more (Supplementary Fig. 3b). There were no consistent patterns with age (Supplementary Fig. 4a, b).

Table 2 and Supplementary Table 5 presents the univariate and multivariable regression results for the number of persistent symptoms >1 month and >3 months. In both univariable models, female sex was associated with a higher rate of persistent symptoms (1.68 times higher (95% CI 1.03, 2.74) for >1 month and 1.95 times higher (95% CI 1.09, 3.45) for >3 months) and the rate did not differ by age group. In the multivariable model for symptom persistence >1 month, both sex (1.68, 95% CI (1.03, 2.73)) and

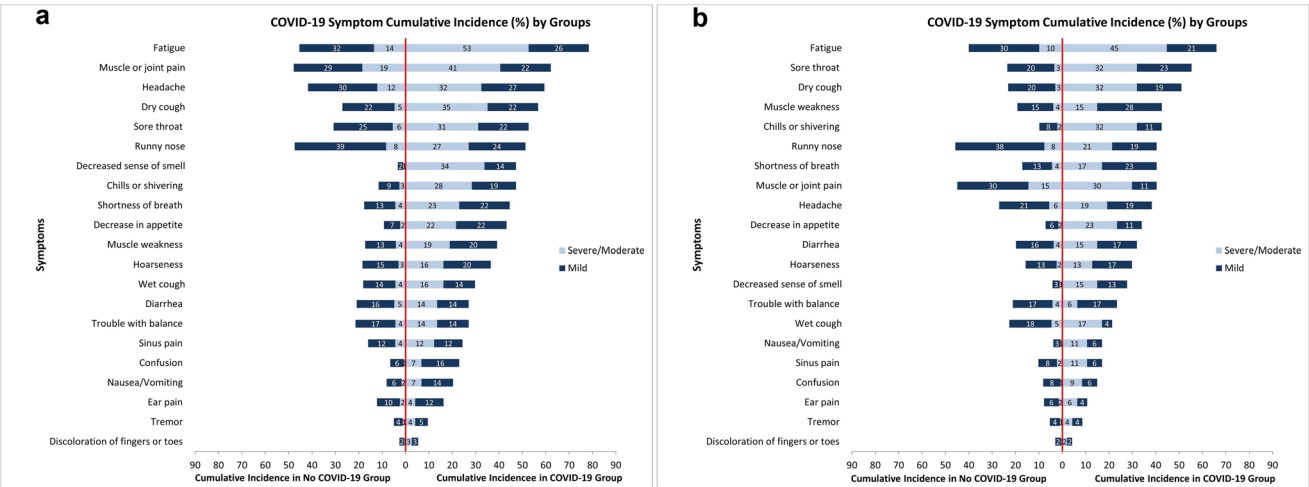

**Fig. 2 Cumulative incidence of mild and moderate/severe symptoms by sex. a** Cumulative incidence percent of mild (dark blue) and moderate/severe (light blue) symptoms since March 2020 reported by non-COVID-19 and COVID-19 female participants. **b** Cumulative incidence percent of mild (dark blue) and moderate/severe (light blue) symptoms since March 2020 reported by non-COVID-19 and COVID-19 male participants. Data to reproduce this figure are included in Supplementary Data 2.

**Fig. 3 Overall cumulative incidence of persistent symptoms.** Overall cumulative incidence percent (light blue) and cumulative incidence percent of symptom persisting for >1 month (dark blue) and >3 months (hatched blue) by severity for the ten most persistent symptoms in CLSA COVID-19 study participants with COVID-19 (*n* = 121). Data to reproduce this figure are included in Supplementary Data 3.

multimorbidity (1.90, 95% CI (1.02, 3.49)) were associated with a greater risk of persistent symptoms. For symptom persistence >3 months, sex was no longer significant in the multivariable model after the addition of subjective social status for which each 1 rung increase was associated with a 15% decrease in the rate of persistent symptoms after adjusting for age, sex and multimorbidity. The estimates from the sensitivity analyses were similar in magnitude and significance (Supplementary Tables 6 and 7).

**Table 2 Univariate and multivariable negative binomial regression analysis results assessing factors associated with >1 month and >3 month symptom persistence in participants with a positive COVID-19 test or physician diagnosis.**

| Factors | Univariate | | Multivariable | |
|---|---|---|---|---|
| | IRR (95% CI) | p value | IRR (95% CI) | p value |
| *Outcome: any persistent symptoms >1 month*[a] | | | | |
| Age group (ref.: <65 years) | | | | |
| 65–74 | 0.75 (0.43, 1.31) | 0.584 | 0.65 (0.38, 1.12) | 0.248 |
| 75+ | 0.81 (0.44, 1.52) | | 0.66 (0.35, 1.25) | |
| Female sex | 1.68 (1.03, 2.74) | 0.037 | 1.68 (1.03, 2.73) | 0.036 |
| *Number of chronic conditions (ref: 0 or 1)* | | | | |
| 2 or more | 1.80 (0.98, 3.23) | 0.053 | 1.90 (1.02, 3.49) | 0.038 |
| *Any depression or anxiety (ref.: No)* | 0.90 (0.54, 1.52) | 0.688 | | |
| *Subjective social status (1 rung)* | 0.95 (0.84, 1.07) | 0.433 | | |
| *Any mobility ADL difficulty (ref.: No)* | 1.40 (0.83, 2.45) | 0.223 | | |
| *Outcome: any persistent symptoms >3 months*[b] | | | | |
| Age group (ref: <65 years) | | | | |
| 65–74 | 1.36 (0.71, 2.59) | 0.561 | 1.11 (0.59, 2.07) | 0.627 |
| 75+ | 0.99 (0.47, 2.11) | | 0.78 (0.37, 1.66) | |
| Female sex | 1.95 (1.09, 3.45) | 0.023 | 1.39 (0.77, 2.51) | 0.260 |
| *Number of chronic conditions (ref: 0 or 1)* | | | | |
| 2 or more | 2.12 (1.03, 4.30) | 0.039 | 1.98 (0.96, 4.09) | 0.065 |
| *Any depression or anxiety (ref.: No)* | 1.15 (0.64, 2.13) | 0.641 | | |
| *Subjective social status (1 rung)* | 0.82 (0.71, 0.95) | 0.004 | 0.85 (0.73, 0.99) | 0.023 |
| *Any mobility ADL difficulty (ref.: No)* | 1.98 (1.10, 3.67) | 0.026 | | |

[a]Sample size for univariate regression models predicting any persistent symptoms >1 month: age group and female sex (n = 118), any depression or anxiety and any mobility ADL difficulty (n = 117), number of chronic conditions (n = 116), subjective social status (n = 114); multivariable model (n = 116).
[b]Sample size for univariate regression models predicting any persistent symptoms >3 months: age group and female sex (n = 116), any depression or anxiety and any mobility ADL difficulty (n = 115), number of chronic conditions (n = 114), subjective social status (n = 112); multivariable model (n = 110).

## Discussion

This is the one of the first studies reporting COVID-19 symptom persistence launched in an existing population-based cohort of older adults, allowing us to estimate not only the background symptom cumulative incidence in the cohort, but also to explore a broader range of pre-pandemic risk factors for post-acute COVID-19 symptom persistence. Whereas other studies have not differentiated between mild and moderate/severe symptoms, we consistently found higher symptom cumulative incidence and severity in the COVID-19 group beyond the background level observed in community-living middle-aged and older adults during the same time frame. In our study of primarily non-hospitalized patients with COVID-19, over two-thirds reported at least one symptom persisting >1 month and over half reported at least one symptom persisting >3 months. This extends previous work and further substantiates long-COVID as an important syndrome in both hospitalized and non-hospitalized individuals.

We demonstrated a higher cumulative incidence and severity of symptoms in participants with COVID-19 compared to those without. Lund et al.[20] was the only other population-based study examining symptom as the time of COVID-19 to include a comparison group. They used information from Danish prescription, patient, and health insurance registries and propensity-score weighting to compare PCR-test positive and negative individuals with respect to hospital-based diagnoses and prescriptions, as well as general practitioner and outpatient clinical visits in 2 weeks to 6 months after diagnosis. While they did not find differences in overall hospital-based diagnoses or prescriptions between those who tested positive and negative, they did find increased prior event adjusted rate ratios for general practitioner and outpatient clinic visits, suggesting that persistent symptoms may lead to increased healthcare utilization but not to the initiation of hospital-based acute treatment.

Most symptom persistence studies to date have been conducted in hospitalized patients. While hospital-based cohorts include the most severe cases of COVID-19, our study provides important evidence for similar levels of persistent symptoms in a population-based cohort where the vast majority of participants with COVID-19, like the general population, did not require hospitalization. In recent systematic reviews, the median prevalence of ≥1 post-acute symptom has been reported as 70.0% (IQR, 46.3–78.9%) by Nalbandian et al.[6] and 72.5% (IQR, 55.0–80.0%) by Nasserie et al.[5]. This is in line with our estimate of 67.8% having at least one symptom persisting >1 month and higher than 53.4% persisting >3 months in community-living individuals with COVID-19. The largest multicenter hospital-based study, PHOSP-COVID, found that only 28.8% of patient were fully recovered at a median of 5.9 months post-discharge[28]. In our study, only 5.8% of individuals with COVID-19 reported any hospitalization. There are fewer studies of non-hospitalized individuals with COVID-19. Hernandez-Romieu et al.[19] reported 68% of participants had a new diagnosis (i.e., a new ICD-10 code) within 1–6 months, Jacobson et al.[13] reported 66.9% of participants had at least one symptom at a median of 4 months, and Gaber reported 45% still had symptoms 3-4 months post COVID-19 diagnosis. While the estimated cumulative incidence of persistent symptoms is slightly lower in primarily non-hospitalized groups, it is nonetheless substantial. The largest community-based study examining 29 persistent symptoms including participants from the REACT-2 study[29] found 37.7% of participants reported symptoms at 12 weeks. Similar to studies in hospitalized and non-hospitalized individuals with COVID-19[5,6,12–14,19,21,28,29], we found the most commonly reported persistent symptoms were fatigue, shortness of breath, dry cough, and muscle and joint pain; however, Matta[21] and Whitaker[29] also reported sleep problems, which we did not capture. We also found that over a third of individuals with COVID-19 who reported moderate/severe shortness of breath, decreased sense of smell, trouble with balance, and wet cough indicated that these symptoms persisted for >3 months, consistent with NICE guidance on long COVID (symptoms lasting >12 weeks)[30]. Importantly,

many of these symptoms can be improved with rehabilitation and these data suggest there may be unmet needs for rehabilitative care among community-dwelling patients with COVID-19 who did not require hospital treatment[31,32].

Finally, we found the rate of persistent symptoms was higher in females, those with pre-pandemic multimorbidity and lower levels of subjective social status but did not differ significantly by age. There have been mixed results on the association between symptom persistence and age with some studies reporting an association with symptom persistence[12] or post-COVID-19 healthcare utilization[19] and others not[13,15]. Although we found differences in the pattern of mild and moderate/severe symptom cumulative incidence by age group qualitatively, we did not find an association between age group and the rate of persistent symptoms in our regression analysis. While the associations with sex[13–15,19,28,29] and multimorbidity[13,15,19,28,29,33] have been previously reported, no studies to date have examined pre-pandemic subjective social status. As many studies have used administrative data or patient records, this type of measure may not be readily available. Until recently no other studies have found an association between an indicator of socioeconomic status and symptom persistence[15,28,29,34]. Interestingly, the two largest studies including community samples in the UK had conflicting results. Thompson et al.[34] conducted a meta-analysis of 10 longitudinal studies and electronic health records. The authors found no significant relationship between a postal-code based index of multiple deprivation in the longitudinal studies but found those in the least deprived areas had increased odds of long-COVID compared to those in the most deprived areas. In contrast, Whitaker et al.[29] found an increased risk of persistent symptoms (>12 weeks) for those living in areas of higher deprivation compared to those living in areas of lower deprivation. Our results using individual-level subjective social status as a predictor of symptom persistence at 3 months support the results of Whitaker et al. Subjective social status has been shown to explain variance beyond more objective measures of SES such as income and education[35]. Studies have also reported an association between race and symptom persistence[13,36] and in post-COVID-19 healthcare utilization[19]. The National Institutes of Health has identified the short- and long-term effects of COVID-19 on health and how to reduce differential outcomes among racial and ethnic groups as a research priority[37]. Our finding that lower subjective social status (reflecting the relative perception that individuals have of their place in the social hierarchy) is associated with COVID-19 symptom persistence may indicate that other social factors should be considered in addition to race in future studies.

This study has many strengths including its design nested within an existing population-based nationally generalizable cohort[23] with pre-morbid data on many aspects of health. This study also has some limitations within which to interpret the results. While the CLSA cohort has been shown to be generalizable to the target population in Canada on many factors[23], the response rate of the COVID-19 study was 67.2% which may lead to participation bias. This is not unusual compared to other studies in non-hospitalized patients that were not based on administrative data, where most response rates were less than 50%[5]. One advantage of using the CLSA data is that we can understand how our COVID-19 questionnaire study population compares to the full CLSA and the target population in Canada, which is not possible for most of the non-hospital-based and hospital-based cohorts published to date. Nonetheless, while our sample size is large, the actual number of participants with COVID-19 was relatively small. While the number of COVID-19 positive participants aligns with Canadian prevalence statistics in community-living older adults for the period in which the Exit

questionnaire was administered[38], the small sample size is reflected in the width of our 95% CIs. Furthermore, by excluding participants reporting they "very likely" had COVID-19 but had no positive COVID-19 test or physician diagnosis, we may have left out some participants who in fact had COVID-19. However, our results were robust to a number of sensitivity analyses. We also do not have the exact date that participants experienced COVID-19. It is possible that some participants had recovered from COVID-19 less than 1 month prior to the questionnaire administration, and thus we may be underestimating the cumulative incidence of symptom persistence. Finally, we cannot comment on symptom persistence longer than 3 months.

One additional potential limitation is that, despite that our COVID-19 prevalence estimates are in-line with other PCR-test-based Canadian data sources at the time, our definition of COVID-19 is based on self-report. Self-reported data are common in population-based community cohorts and while they may be more susceptible to recall bias, a positive COVID-19 test or physician diagnosis was such a salient event in the first year of the pandemic that recall bias is likely be minimal. Furthermore, measuring COVID-19 antibody status retrospectively also has limitations. Matta et al. examined the association between antibody-based results and people's self-reported belief that they had COVID-19 with persistent symptoms in the CONSTANCES cohort. Because that author found significantly larger associations with persistent symptoms with belief compared to antibody results they concluded that "physical symptoms persisting 10-12 months after the COVID-19 pandemic first wave may be associated more with the belief in having experienced COVID-19 infection than with actually being infected with the SARS-CoV-2 virus". This may be problematic however because their conclusions were based on measuring antibody status using dry blood spots collected up to 8 months post COVID-19 infection. The authors did not consider the issue of false positives or the issue of decaying antibodies with time since infection which could decrease the presumed specificity of their assay[39]. Even with a reported specificity of 97.5% the number of false positives can be substantial with a low cumulative incidence condition in a large population-based study. While the authors note in the discussion that the negative predictive value of the test in quite high (99.4%) they do not take into account that the positive predictive value of the assay is only 59.2%. This is not to say that population-based seroprevalence studies are not important or reliable, but rather that the totality of evidence needs to be considered to move our understanding forward. With the new omicron variant, many countries have substantially reduced PCR testing. Self-reported COVID-19 status and symptoms may become increasingly salient. Evidence from large well-established cohorts will allow us to examine factors related to health and aging pre- during- and eventually post-pandemic.

**Clinical and public health implications**. All follow-up studies of hospitalized COVID-19 patients post acute-care that incorporated assessments of health-related quality of life and functional capacity measures have universally reported substantial deficits in these domains[6]. However, because most COVID-19 infected individuals are managed in the community[40], it is of major public health importance to better understand the longer-term consequences of COVID-19 in the general population. With millions of individuals experiencing COVID-19 illness, persistent symptoms are a burden on individuals and their families as well as on outpatient care and public health. We found that a non-trivial proportion of people living in the community who may not have been hospitalized for COVID-19 still experience symptoms 1 month and even 3 months post infection. Many of the

symptoms, for example, shortness of breath, fatigue, and pain, are amenable to rehabilitation. These data support the WHO recommendation that rehabilitation has an important role to play in promoting recovery after COVID-19[31].

## Data availability

Data are available from the Canadian Longitudinal Study on Aging for researchers who meet the criteria for access to de-identified CLSA data. Public sector researchers from institutions that can provide research ethics board approval can email access@clsa-elcv.ca to request a Magnolia user account to apply for CLSA data. The source data needed to reproduce Figs. 1, 2a, b, and 3 can be found in Supplementary Data 1, 2, and 3, respectively.

## Code availability

No custom code was used in the analyses of these data. R and SAS syntax that supports the results of this study can be made available upon request from the corresponding author.

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

## Acknowledgements

Funding for the support of the CLSA COVID-19 Questionnaire based study is provided by Juravinski Research Institute, Faculty of Health Sciences, McMaster University, Provost Fund from McMaster University, McMaster Institute for Research on Aging, Public Health Agency of Canada and the Nova Scotia COVID-19 Health Research Coalition. Funding for the Canadian Longitudinal Study on Aging (CLSA) is provided by the Government of Canada through the Canadian Institutes of Health Research (CIHR) under grant reference: LSA 94473 and the Canada Foundation for Innovation. This research has been conducted using the CLSA COVID-19 Data Version 0, CLSA Baseline Data Version 3.5 (tracking cohort) and 4.2 (comprehensive cohort) and CLSA follow-up 1 Data Version: 2.1 (tracking cohort) and 3.0 (comprehensive cohort). The CLSA is led by P.R., C.W. and S.K. P.R. holds the Raymond and Margaret Labarge Chair in Optimal Aging and Knowledge Application for Optimal Aging, is the Director of the McMaster Institute for Research on Aging and the Labarge Centre for Mobility in Aging, and holds a Tier 1 Canada Research Chair in Geroscience. L.E.G. is supported by the McLaughlin Foundation Professorship in Population and Public Health. N.E.B. holds a Tier 2 Canada Research Chair in Infectious Disease Prevention.

## Author contributions

L.E.G., J.M., C.W., S.K., and N.E.B. designed the CLSA COVID-19 study. L.E.G., M.B., J.M., and P.R. contributed to the study design and analysis plan. S.B. and U.E.O. conducted the data analyses with consultation from M.T. L.E.G. wrote the manuscript with input from all authors. The members of the CLSA team have contributed to the collection of the data across Canada. The opinions expressed in this manuscript are the authors' own and do not reflect the views of the Canadian Longitudinal Study on Aging.

## Competing interests

The authors declare no competing interests.

## Additional information

---

## on behalf of the Canadian Longitudinal Study on Aging (CLSA) Team

Laura Anderson[1], Cynthia Balion[11], Andrew Costa[1], Yukiko Asada[9], Benoît Cossette[12], Melanie Levasseur[12,13], Scott Hofer[14], Theone Paterson[14], David Hogan[5], Teresa Liu-Ambrose[15], Verena Menec[16], Philip St. John[16,17], Gerald Mugford[18], Zhiwei Gao[18], Vanessa Taler[19], Patrick Davidson[19], Andrew Wister[20] & Theodore Cosco[20]

[11]Department of Pathology and Molecular Medicine, McMaster University, Hamilton, ON, Canada. [12]Department of Community Health Sciences, University of Sherbrooke, Sherbrooke, QC, Canada. [13]School of Readaptation, University of Sherbrooke, Sherbrooke, QC, Canada. [14]Department of Psychology, University of Victoria, Victoria, BC, Canada. [15]Physical Therapy, University of British Columbia, Vancouver, BC, Canada. [16]Department of Community Medicine, University of Manitoba, Winnipeg, MB, Canada. [17]Department of Geriatric Medicine, University of Manitoba, Winnipeg, MB, Canada. [18]Faculty of Medicine, Memorial University of Newfoundland, St. John's, NL, Canada. [19]School of Psychology, University of Ottawa, Ottawa, ON, Canada. [20]Department of Gerontology, and Gerontology Research Centre, Simon Fraser University, Vancouver, BC, Canada.

