## [Peer Review File · Communications Medicine]

Reviewers' comments:

Reviewer #1 (Remarks to the Author):

I have thoroughly assessed the answers given to the statistics and public health reviewers. The authors are honest in their reporting of limitations and there are no further comments to raise.

Reviewer #2 (Remarks to the Author):

I think that previous comments have been satisfactorily addressed. However, if feasible, it may be useful to do another sensitivity analysis using a weighting approach such as direct standardisation to produce estimates that are as they would have been if the population has the distribution of the census data in terms of key variables.

Reviewer #3 (Remarks to the Author):

The authors were very responsive to the peer reviews. While the number of people with COVID is small relative to all those in the study, this appears to be because the data were collected early in the pandemic before there were many infections. While more infections would have of course provided more information, it also likely would have introduced more potential for misclassification among other complicating factors. So this is a strength of the study in my view, not a weakness. Much information on the occurrence of persistent symptoms in both those who became infected and those who did not is also a very valuable contribution to the literature, as is the community-based nature of the cohort. However, it also seems possible that there could have been some before-after comparisons of symptoms and possibly IADLs in those who had COVID as compared with those who did not (i.e, individuals as their own controls), which would add to the already valuable information presented.

Given the response rates of the COVID sub-study, which the authors have adequately contextualized, it would be important list the potential for selection/participation bias as a limitation of their study.

Would there be value in presenting age-and sex-adjusted estimates for Figures 1-3. Age and sex could be confounding these crude comparisons of symptom-specific persistence between COVID and non-COVID groups, as well as severe/non-severe comparisons in Fig 3. It is hard to see some of the numbers in the dark blue sections of the bars.

It does not appear that brain fog/confusion was measured. This should be listed as a limitation, as it is a common long COVID symptom.

Some comments to help make the manuscript clearer:

Abstract (methods): cumulative incidence (used in results) not defined, including over what

period?

Abstract (results, first sentence): What is the value of reporting combined cumulative incidence estimates when the goal is to compare those with and without COVID? Are these estimates after COVID-19 status was assessed? Or does it include data from the prior study waves? The timing of all of the estimates in the abstract, especially in relation to COVID onset, was unclear to me and should be clarified in the abstract. Perhaps in the 'Outcome' section. This all becomes clearer in the paper, but it is not clear in the abstract.

Abstract (results): I am fine with the use of objective social status, as it appears well-justified by the authors. But most readers of the abstract will not be able to discern what a '1 unit increase in subjective social status' might mean without more context. One way to fix this for the abstract would just be to say in the abstract that the rate of symptoms was lower among those with higher social status. Readers can see the details and effect estimates in the paper.

Methods

Lines 123 and 126. The authors should distinguish between the CLSA baseline questionnaire and the baseline interview for the COVID sub-study.

Exclusion of the 357 participants. The serologic data help provide further justification. However, details on 'Exclusions' should probably go in the 'Participants' paragraph (Line 130) as opposed to the 'Variables' section. And it is worth exploring/discussing the potential for bias in excluding these individuals, some of whom may have had COVID.

Responses to Referees' Comments:

Reviewer #1: Medical Statistics and Epidemiology

Remarks to the Author:

- 1. I have thoroughly assessed the answers given to the statistics and public health reviewers. The authors are honest in their reporting of limitations and there are no further comments to raise.**

Response: Thank you for reviewing the manuscript.

Reviewer #2: Medical Statistics and Epidemiology

Remarks to the Author: I think that previous comments have been satisfactorily addressed. However, if feasible, it may be useful to do another sensitivity analysis using a weighting approach such as direct standardisation to produce estimates that are as they would have been if the population has the distribution of the census data in terms of key variables.

Response: Thank you for this comment. We also were concerned that the COVID-19 and non-COVID-19 groups differed in terms of sociodemographic characteristics. As well we know that multimorbidity is associated with both symptoms and was more common in those with COVID-19. As a sensitivity analysis we chose to do a direct standardization in the COVID-19 group to reflect the age-, sex-, and multimorbidity-distribution of the non-COVID-19 group. We present the results in e-figure 1 and found that the symptoms in the COVID-19 group were only slightly attenuated. Given that the COVID-19 and non-COVID-19 groups did not differ significantly with respect to age and sex (table 1), and we do not have access to population-level distribution statistics on key variables other than age and sex (e.g., multimorbidity), we were doubtful that such a sensitivity analysis would provide additional insights beyond what has already been done. For these reasons we did not conduct the additional sensitivity analysis.

Reviewer #3: Public Health, COVID-19, Long COVID

Remarks to the Author:

- 1. The authors were very responsive to the peer reviews. While the number of people with COVID is small relative to all those in the study, this appears to be because the data were collected early in the pandemic before there were many infections. While more infections would have of course provided more information, it also likely would have introduced more potential for misclassification among other complicating factors. So this is a strength of the study in my view, not a weakness.**

Response: Thank you for your comments. We have updated the limitations to underscore that the number of COVID-19 positive cases is in-line with the prevalence in community-living older adults early in the pandemic (discussion, page 20, paragraph 1):

“While the number of COVID-19 positive participants aligns with Canadian prevalence statistics in community-living older adults for the period in which the Exit questionnaire was administered[38], the small sample size is reflected in the width of our 95% confidence intervals.”

- 2. Much information on the occurrence of persistent symptoms in both those who became infected and those who did not is also a very valuable contribution to the literature, as is the community-based nature of the cohort. However, it also seems possible that there could have been some before-after comparisons of symptoms and possibly IADLs in those who had COVID as compared with those who did not (i.e, individuals as their own controls), which would add to the already valuable information presented.**

Response: We agree that within-person longitudinal data would have been of interest, but symptom data were not available prior to the pandemic. Of interest, a paper using CLSA data led by Dr. Marla Beauchamp (doi:10.1001/jamanetworkopen.2021.46168) examined functional mobility pre-pandemic (2015-2018) and during the pandemic collected at the COVID-19 exit questionnaire. Participants were asked about the degree of difficulty in standing up after sitting in a chair, walking up and down a flight of stairs without assistance, and walking 2 to 3 neighborhood blocks at both time points. We found that COVID-19 status was significantly associated with worse self-reported changes in mobility and a decline in functioning outcomes even in the absence of hospitalization.

- 3. Given the response rates of the COVID sub-study, which the authors have adequately contextualized, it would be important list the potential for selection/participation bias as a limitation of their study**

Response: Thank you for this comment. Our response rate of 67.2% was relatively high compared to other persistent symptom studies in non-hospitalized patients. Of the studies described in our introduction (that were not based on administrative data), 25% did not report a response rate, and of those that did, the median was 50%. One of the big benefits of using the CLSA data is that we can understand how our COVID-19 questionnaire study population compares to the full CLSA and the target population in Canada, which is not possible for most of the non-hospital based and hospital-based cohorts.

As previously noted, our self-reported COVID-19 cumulative incidence rates are in-line with those reported by the Public Health Agency of Canada. As well, in a paper led by Dr. Nicole Basta, we found that self-reported vaccine willingness in the first year of the pandemic aligned closely with COVID-19 vaccination uptake in Canada (assessed Jan 22, 2022) by age group (<https://doi.org/10.1093/aje/kwac029>). We do, however, agree that participation bias is still possible. We have clarified the potential for participation bias in the limitations section (discussion, page 19, paragraph 2):

*“While the CLSA cohort has been shown to be generalizable to the target population in Canada on many factors[23], the response rate of the COVID-19 study was 67.2% **which may lead to participation bias.**”*

4. Would there be value in presenting age- and sex-adjusted estimates for Figures 1-3. Age and sex could be confounding these crude comparisons of symptom-specific persistence between COVID and non-COVID groups, as well as severe/non-severe comparisons in Fig 3. It is hard to see some of the numbers in the dark blue sections of the bars.

Response: We also were concerned that the COVID-19 and non-COVID-19 groups may have differed in terms of sociodemographic characteristics. To better understand confounding and effect modification by age and sex for both the prevalence of individual symptoms in the COVID and non-COVID groups and for persistent symptoms in the COVID group, we investigated several things:

- We conducted a direct standardization in the COVID-19 group to reflect the age-, sex-, and multimorbidity-distribution of the non-COVID-19 group as a sensitivity analysis (as we know multimorbidity is also associated with both symptoms and is more common in those with COVID-19). We present the results in e-Figure 1 and found that the symptoms in the COVID-19 group were only slightly attenuated.
- We conducted sex- and age- stratified analyses for the cumulative incidence symptoms analysis. We found that female participants with COVID-19 tended to report more moderate/severe symptoms than males (Figure 2a) and that participants with COVID-19 who were <65 years old reported a higher cumulative incidence of symptoms than those 65-74 and 75+ years, however a higher proportion of the symptoms reported in the 75+ COVID-19 group were moderate/severe (eFigure 2a-2c). ‘
- We also conducted sex- and age-stratified analyses when examining symptom persistence in the COVID-19 group. We found approximately 60% of male participants reported at least one symptom persisting >1 month compared to 75% of female participants, with over 10% of females reporting 8 or more persistent symptoms compared to 4% of males (eFigure S3a); and more than 50% of male and 40% of female participants reported no persistent symptoms >3 months, while over 10% of females compared to 2% of males reported 5 or more (eFigure S3b). We did not find consistent

patterns with age (Figure S4a-b).

To improve readability, we have changed the color from dark blue to light red for all symptom cumulative incidence figures (Figure 1, Figure 2a-b, eFigure 1, eFigure 2a-c).

5. It does not appear that brain fog/confusion was measured. This should be listed as a limitation, as it is a common long COVID symptom.

Response: Confusion was included in the list of potential symptoms. Confusion was reported by 19.8% of the COVID-19 group and 7.3% of the non-COVID-19 group (Figure 1). To improve the readability of our figures we have increased the font size for the symptoms cumulative incidence figures.

Some comments to help make the manuscript clearer:

6. Abstract (methods): cumulative incidence (used in results) not defined, including over what period?

Response: We have indicated the period at the end of the of the methods section of the abstract:

“We used data from a cross-sectional COVID-19 questionnaire administered from September-December 2020 linked to baseline (2011-2015) and follow-up 1 (2015-2018) data from a pre-existing population-based cohort including 23,757 adults 50+ years from 10 Canadian provinces to examine how age, sex, and indicators of pre-pandemic physical, psychological, social, and function were related to the severity and persistence of 23 COVID-19-related symptoms experienced between March 1st 2020 and COVID-19 questionnaire completion.”

7. Abstract (results, first sentence): What is the value of reporting combined cumulative incidence estimates when the goal is to compare those with and without COVID? Are these estimates after COVID-19 status was assessed? Or does it include data from the prior study waves? The timing of all of the estimates in the abstract, especially in relation to COVID onset, was unclear to me and should be clarified in the abstract. Perhaps in the 'Outcome' section. This all becomes clearer in the paper, but it is not clear in the abstract.

Response: Our intention was to underscore that while many COVID-related symptoms were also reported by people that did not experience COVID-19, people with COVID-19 experienced far more moderate/severe symptoms. Many studies to date report only symptoms experienced by those with COVID-19. One of the advantages of our study is that we can describe the symptoms

reported in the first year of the pandemic in community-living older adults who had COVID-19 compared to those who did not. We have made the change noted in the response #6 to better describe the timing of the symptoms. We have also re-written the first two sentences of the abstract to better describe the timing of COVID onset and to more clearly make this point:

“The most common symptoms during the first year of the pandemic were fatigue, dry cough, muscle/joint pain, sore throat, headache, and runny nose; reported by over 25% of participant who had (n=121) or did not have (n=23,636) COVID-19 during the study period. The cumulative incidence of moderate/severe symptoms in people with COVID-19 was more than double that reported by people without COVID-19, with the absolute difference ranging from 16.8% (runny nose) to 37.8% (fatigue).”

8. Methods: Lines 123 and 126. The authors should distinguish between the CLSA baseline questionnaire and the baseline interview for the COVID sub-study.

Response: We have updated the text in the Study design/Setting section to distinguish between the CLSA baseline questionnaire and the COVID-19 baseline questionnaire (Methods, page 6, paragraph 1):

*“Recruitment and **CLSA** baseline data were collected on 51,338 participants in 2011-2015, follow-up 1 (FUP1) was completed on 48,893 participants (95% retention) in 2015-2018, and FUP2 was completed in mid-2021. The CLSA COVID-19 study was launched on April 15, 2020 and comprised a **COVID-19** baseline questionnaire, weekly/biweekly/monthly symptoms questionnaires, and an Exit questionnaire.”*

9. Methods: Exclusion of the 357 participants. The serologic data help provide further justification. However, details on 'Exclusions' should probably go in the 'Participants' paragraph (Line 130) as opposed to the 'Variables' section. And it is worth exploring/discussing the potential for bias in excluding these individuals, some of whom may have had COVID.

Response: We have moved the description of the exclusion of the 357 participants to the 'Participants' section. It is difficult to speculate how the exclusion of participants indicating that it was “very likely” they had COVID-19 but did not have a positive COVID-19 test or physician diagnosis may have biased our results. Given the number of people without COVID-19 who experienced COVID-19-related symptoms (Figure 1) and the results of our seroprevalence study, it is unlikely that all of the 357 participants had COVID-19, which means we would be adding a significant number of false positives to our COVID-19 group. It is plausible that the 357 participants could have more symptoms (or more severe symptoms) and felt they did not need additional verification from a physician and/or COVID-19 PCR test. It is also plausible that they had fewer symptoms and did not feel the need to seek health care due to their presumed

infection. Additionally, it is plausible that they were similar to the included “COVID-19” group and were not tested due to limited access to testing and health care during some periods of the pandemic. We tried to examine the possible impact of adding those for whom we had the most evidence that they experienced COVID-19 based on their seroprevalence results as a sensitivity analysis and it did not substantially impact our results. We have added this as a potential limitation in the discussion (page 20, paragraph 1):

“Furthermore, by excluding participants reporting they “very likely” had COVID-19 but had no positive COVID-19 test or physician diagnosis, we may have left out some participants who in fact had COVID-19. However, our results were robust to a number of sensitivity analyses.”

REVIEWERS' COMMENTS:

Reviewer #2 (Remarks to the Author):

Thank you for addressing previous comments. I have no further comments.

Reviewer #3 (Remarks to the Author):

The authors have been adequately responsive to my comments/suggestions/queries.